# Enhanced Experts with Uncertainty-Aware Routing for Multimodal Sentiment Analysis

## ABSTRACT

Multimodal sentiment analysis, which has garnered widespread attention in recent years, aims to predict human emotional states using multimodal data. Previous studies have primarily focused on enhancing multimodal fusion and integrating information across different modalities, while overlooking the impact of noisy data on the internal features of each single modality. In this paper, we propose the **E**nhanced experts with **U**ncertainty-**A**ware **R**outing (**EUAR**) method to address the influence of noisy data on multimodal sentiment analysis by capturing uncertainty and dynamically altering the network. Specifically, we introduce the Mixture of Experts approach into multimodal sentiment analysis for the first time, leveraging its properties under conditional computation to dynamically alter the network in response to different types of noisy data. Particularly, we refine the experts within the MoE framework to capture uncertainty in the data and extract clearer features. Additionally, a novel routing mechanism is introduced. Through our proposed U-loss, which utilizes the quantified uncertainty by experts, the network learns to route different samples to experts with lower uncertainty for processing, thus obtaining clearer, noise-free features. Experimental results demonstrate that our method achieves state-of-the-art performance on three widely used multimodal sentiment analysis datasets. Moreover, experiments on noisy datasets show that our approach outperforms existing methods in handling noisy data. Our anonymous implementation code can be available at https://anonymous.4open.science/r/EUAR-7BF6.

## CCS CONCEPTS

• **Information systems** → **Sentiment analysis**; • **Computing methodologies** → *Artificial intelligence*.

## KEYWORDS

Multimodal Sentiment Analysis, Uncertainty Learning, Mixture of Experts

## 1 INTRODUCTION

In today's digital era, the collective influence of multiple modalities has garnered widespread attention, with multimodal learning emerging as one of the current research hotspots. Among these, Multimodal Sentiment Analysis (MSA), as a core task in multimodal learning, has also received significant interest from researchers.

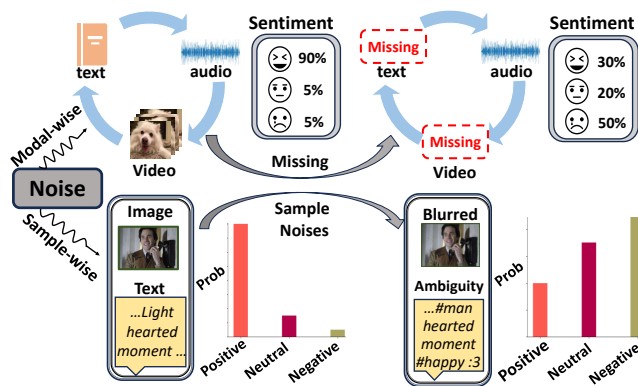

**Figure 1: An illustrative example of the two types of noise in MSA task: modal-wise noise and sample-wise noise. The former may occur due to the malfunction of perceptive devices and result in corrupted classification results. Meanwhile, the latter exists in the samples such as blurred images, ambiguous text, and so on, which produce low quality data and disrupt model's prediction tremendously.**

Most existing methods [1–4] approach this task from the perspective of multimodal fusion, addressing the interaction of information across modalities at different levels such as early [5], intermediate [6, 7], and late fusion [8], on raw data, feature, and decision levels respectively. Their efforts primarily focus on resolving information interaction among modalities, and these works have extensively demonstrated the effectiveness of fusion in the MSA task.

However, prevalent approaches to multimodal sentiment analysis exhibit a notable limitation: *they tend to overly prioritize intermodal information interaction, often neglecting the presence of noise within individual modalities and the diverse degrees of noise across different samples.* We comprehend these two types of noise as modal-wise and sample-wise. Put simply, when confronted with noisy data, existing networks inadequately address this challenge. The processing of multimodal data inevitably entails encountering noise scenarios such as modality missing, image blurriness, and text ambiguity. As depicted in Figure 1, scenarios such as audio interruptions, image distortions, and text ambiguities lead to a scenario where samples with ground truth as positive are misclassified by most networks as neutral or even negative. These noise instances can result in catastrophic consequences for network performance.

Recently, some efforts [7, 9–11] have begun to address this issue by contributing solutions to modalities' noise scenarios like missing and blurriness. For instance, a recent work [9] quantifies the uncertainty level of the model regarding different modality data noise and incorporates uncertainty-weighted fusion. In [7], Wang *et al.* work reconstructs data samples using multimodal flows to address modality missing scenarios. Although these methods have

somewhat mitigated the impact of noise on networks, their performance still fall short when facing samples suffering from varying degrees of noise.

Therefore, based on these challenges, we pose two questions: *(1) How to quantify noise in samples to obtain clearer features?* Inspired by [12], we map different sample instances to a high-dimensional Gaussian distribution space and utilize sample variance to quantify inherent uncertainty. *(2) How to dynamically adapt our network according to different noise scenarios?* We introduce a Mixture of Experts (MoE) and utilize conditional computation to dynamically alter our network based on the noise scenario of different samples. Based on these considerations, we propose a novel MSA model in this paper, termed *Enhanced experts with Uncertainty-Aware Routing (EUAR)*, addressing the shortcomings of previous works.

Specifically, our work can be comprised of the following components: (1) Mapping samples to multimensional Gaussian distributions using different experts in EUAR, to quantify uncertainty using variance, and generate noise-free features using means. (2) Introducing the MoE structure to dynamically select different experts based on different samples, dynamically altering the network structure according to the scenario of different samples. (3) Training on basis of uncertainty-based dynamic routing algorithms, which enable the model to select experts with more confidence for specific samples to handle data.

Overall, our contributions can be summarized as follows:

- To our best knowledge, we are the first to introduce the MoE structure into multimodal sentiment analysis, dynamically altering the network to address noise within modalities and varying degrees of noise across different samples, thus complementing the shortcomings of previous works.
- We devise a novel routing algorithm to enhance the experts, enabling the network to quantify uncertainty regarding noise and utilize this uncertainty to guide the routing algorithm in selecting experts with more confidence to process samples.
- We conduct extensive experiments on three widely used multimodal sentiment analysis datasets, achieving state-of-the-art performance and demonstrating superior performance under noise conditions compared to existing methods.

## 2 RELATED WORKS

### 2.1 Multimodal Sentiment Analysis

Previous research aimed to find superior multimodal fusion methods to comprehensively integrate multimodal information for MSA task. These methods mainly focus on early fusion, feature fusion, late fusion, etc. For instance, graph fusion [13] based on feature fusion applies graph learning to integrate features, promoting fusion through constructing different nodes and integrating multimodal information. [9] employed a late fusion approach, merging decision levels by acquiring different weight proportions. These methods have achieved notable success in multimodal sentiment analysis. However, there are still some unsolved problems: both in video datasets and text-image datasets, the varying quality of video frames, images, and text poses challenges for these traditional methods. When encountering situations with low video quality, blurry images, and ambiguous text, the performance of these methods significantly diminishes, making it challenging to accurately identify the sentiment represented in the data. Recently, some work [14, 15] theoretically confirmed that noise in the data can bottleneck the model's performance. To address this issue, we propose the EUAR method, dynamically altering the network to effectively alleviate the bottleneck caused by data noise,i.e., modal-wise and sample-wise noise in sentiment analysis.

### 2.2 Uncertainty in Deep Learning

In general, uncertainty in deep learning can be categorized into epistemic uncertainty and aleatoric uncertainty. The former represents the uncertainty inherent in the neural network itself, indicating the confidence level in its predictions. The latter refers to the inherent noise present in the data, such as image blurriness, text ambiguity, or missing modalities, which pose significant challenges to the model's predictions. In recent years,some efforts have been made to address uncertainty in conventional tasks like face recognition [12],image classification [16] and semantic segmentation [17]. It is worth noting that such uncertainty is also prevalent in multimodal sentiment analysis which impose negative impacts on final decisions. Due to the need for integrating multimodal data, if one modality is corrupted by noise, the overall multimodal features are correspondingly viciously affected. Recently, some research [7, 9, 10] has focused on addressing uncertainty in multimodal learning. However, these efforts primarily concentrate on integrating deterministic information across different modalities without considering the uncertainty within individual modalities. Therefore, we develop this work to fill this gap, aiming to overcome the information bottleneck caused by data noise in the MSA task.

### 2.3 Mixture of Experts

Mixture of Experts, a technique for expanding model parameters, has attached widespread attention in recent years alongside the rise of large models. It has been extensively explored in the fields of computer vision [18–20] and natural language processing [21–23]. MoE employs a learned gating mechanism to selectively activate different experts, adapting the network dynamically by activating k experts tailored to handle specific inputs. This dynamically alters the network without increasing additional computational overhead, thereby expanding the model's capacity. However, current research [24–26] has primarily focused on utilizing MoE to augment model parameters, neglecting the superiority of dynamic networks themselves in the context of conditional computation. Conditional computation [19, 22] selectively activates relevant parts of the model based on input-dependent factors, dividing the model into smaller, specialized sub-models. Different experts exhibit targeted effects for specific inputs. Therefore, we introduce MoE for the first time into the task of multimodal sentiment analysis, enhancing MoE's experts and designing an uncertainty-guided gating mechanism. This enables experts to capture uncertainty and dynamically selects corresponding experts based on the varying levels of noise in different inputs, addressing challenges posed by noise in multimodal sentiment analysis.

## 3 PROPOSED METHODS

Our proposed EUAR method is a multimodal sentiment analysis framework that dynamically adapts its network structure based on

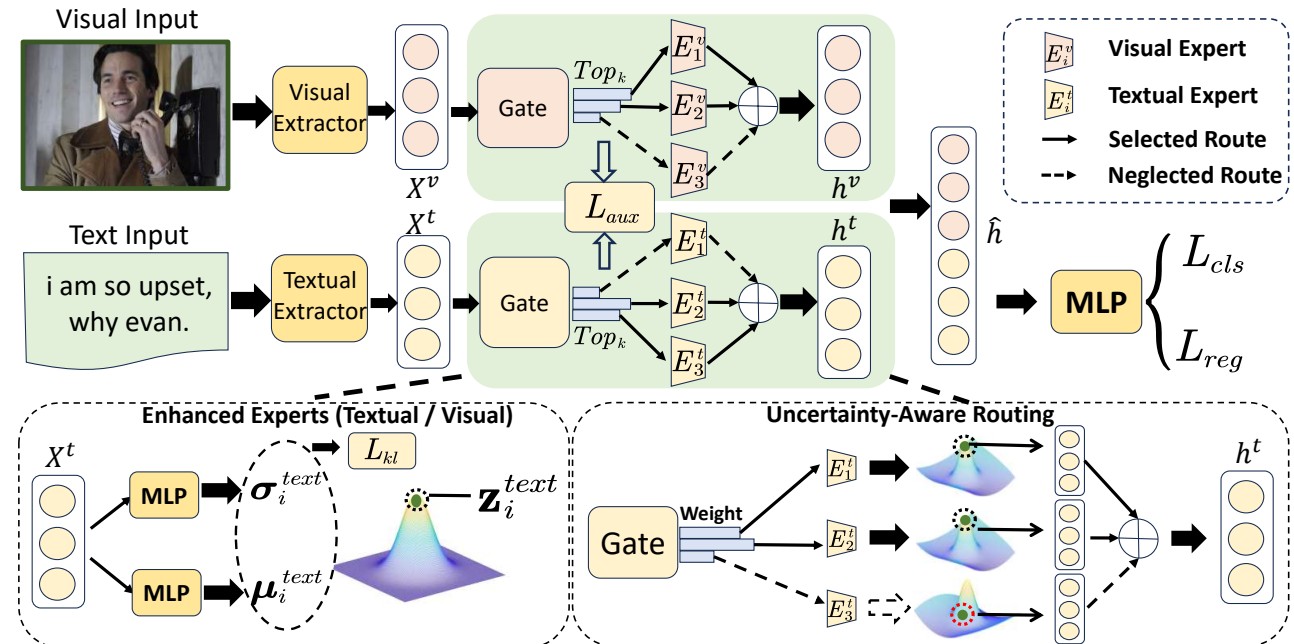

**Figure 2: An illustration of our proposed EUAR method. It encompasses the MoE framework we introduced, along with a comprehensive delineation of the expert components.**

different inputs. Our task is to predict the corresponding sentiment intensity values using given video clips or image-text pairs. Taking image-text pairs as an example, in one input sample, there are two different modal data:textual, and visual.

As shown in Figure 2, we first perform feature extraction in a specific manner and then use the extracted features in our proposed EUAR to predict the corresponding sentiment intensity labels.

## 3.1 Feature Extraction

To align with prior research [1, 9] for fair comparison, we employed FACET, COVAREP, and BERT as feature extractors for the visual, audio, and text modalities, respectively, on the tri-modal dataset. For the bi-modal dataset, ResNet-152 and BERT are utilized as feature extractors for the visual and text modalities, respectively. The following narrative in this paper will be based on the tri-modal dataset. Given video clips, we feed the visual($v$), audio($a$), and text($l$) modalities into their respective feature extractors to obtain samples $X_m^{B \times D}, m \in \{v, a, l\}$ required for this task, where $B$ represents batch size and $D$ represents feature dimensions. The objective is to integrate the features $v$, $a$, and $l$, and predict continuous sentiment intensity values $y \in R$.

## 3.2 Enhanced Experts

In MoE technique, experts refer to a group of sub-models or neural networks, each responsible for processing different aspects or sub-spaces of the input data. These experts can be seen as "specialists" focusing on different tasks or data distributions. In previous works, experts were often defined as multi-layer perceptrons. Inspired by [12], we enhanced the experts to capture uncertainty when dealing with samples influenced by noise to varying degrees. After feature

extraction, we obtain multimodal features $X_i^m, m \in \{v, a, l\}$,where i represents the instance. We map the features of each modality to a diagonal multivariate normal. Specifically, we define the representation $z_i^m$ of each sample $x_i^m$ in the latent space as a Gaussian distribution,which can be represented as:

$$p(z_i^m|x_i^m) \sim N(\mu_i^m, \sigma_i^{m2}I), \tag{1}$$

$$\mu_i^m = f_{\theta_1^m}, \sigma_i^m = f_{\theta_2^m}, \tag{2}$$

where $m$ represents different modalities, and $\theta$ represents the parameters in the neural network. For the parameters $\mu$ and $\sigma$ of the Gaussian distribution, we employ a neural network to predict, where $f_1$ and $f_2$ respectively denote one fully connected layer, with non-shared weights. The feature representation of each modal sample is no longer a fixed value, but sampled from the distribution. Here, $\mu_i^m$ represents the features closer to the true representation under noise-free conditions, while $\sigma_i^m$ represents the aleatoric uncertainty of the modality data. However, the gradients cannot be back-propagated after sampling, thus we employ the re-parameterization trick:

$$z_i^m = \mu_i^m + \epsilon \sigma_i^m, \epsilon \sim N(0, I). \tag{3}$$

Here, Eq. (3) can be interpreted as stable sample representation $\mu_i^m$ being perturbed by noise $\sigma_i^m$ to obtain$z_i^m$. During training, the loss function of the ultimate task will mitigate the impact of this perturbation, leading to $\sigma_i^m$ being learned to be very small. This results in $z_i^m$ being almost identical to $\mu_i^m$, thus losing its ability to quantify uncertainty. Therefore, we adopt a prior method [12] and introduce a $KL$ divergence regularization term:

$$L_{kl}^m = KL[N(z_i^m|\mu_i^m, \sigma_i^{m2})||N(\epsilon|0, I)]$$
$$= -\frac{1}{2}(1 + log\sigma_i^{m2} - \mu_i^{m2} - \sigma_i^{m2}). \tag{4}$$

We constrain $N(\mu_i^m, \sigma_i^{m2})$ to be close to $N(0, I)$ so that $\sigma_i^{m2}$ can accurately quantify the magnitude of uncertainty.

## 3.3 Adaptive MoE Architecture

In the preceding section, we defined the concept of an "expert" and endowed it with the capability to capture uncertainty, allowing us to quantify the magnitude of uncertainty in noisy data based on the value of $\sigma^2$. Next, we will integrate it into the MoE framework. The MoE consists of two main components: the Expert and the Gate. The expert is responsible for processing each sample effectively, while the Gate determines which expert is best suited to handle a particular sample. We denote these two components as $E$ and $G$, respectively. Given the feature $x_i^m$ as input, and the gate selects the top k best experts, preserving the corresponding processing results $E$ for further steps. The algorithm for routing by the gate can be expressed as:

$$G(x_i^m) = TOP_k(softmax(Linear(x_i^m))), \quad (5)$$

where the output dimension of the Linear layer is equal to the number of experts. The $TOP_k$ operation involves setting all values except the top k values to zero. We select the top k experts corresponding to the highest $softmax$ output scores, and use their scores as weights for the expert output results. When the total number of experts is $N$, this process can be formalized as:

$$h_i^m = \sum_j^N G_j^m(x_i^m) E_j^m(x_i^m). \quad (6)$$

However, during training, routing algorithms often tend to favor a few specific experts, rendering the remaining experts ineffective. This leads to the degradation of the entire framework into a static network, unable to dynamically adapt to different noise. To tackle this issue, we adopt a similar approach as in the Switch Transformer [23], incorporating the following auxiliary loss:

$$L_{aux}^m = \frac{1}{N} \sum_j^N R_j^m P_j^m, \quad (7)$$

where $R_j^m$ represents the proportion of samples allocated to the expert $j$,

$$R_j^m = \frac{1}{B} \sum_{x \in B} 1\{argmax G^m(x^m) = j\}. \quad (8)$$

Moreover, $G_j^m$ represents the proportion of weights allocated by the router to expert $j$, and it is defined as:

$$P_j^m = \frac{1}{B} \sum_{x \in B} G_j^m(x^m). \quad (9)$$

Through this regularization constraint, we enforce a similar number of samples allocated to each expert during training, and ensure that $G^m$ yields weights of similar magnitudes for each expert.

## 3.4 Uncertainty-Aware Routing

So far, we have obtained a dynamic network capable of capturing uncertainty. Next, we will empower the Gate the ability to choose experts based on uncertainty. Specifically, we use the magnitude of the $\sigma_i^{m2}$ value to quantify the uncertainty in noisy data. A larger $\sigma_i^{m2}$ value indicates greater uncertainty, while a smaller $\sigma_i^{m2}$ indicates lesser uncertainty. During a forward pass, each expert generates a $\sigma_i^m$ to express its uncertainty in processing the data. We aim for the Gate to select experts with lower uncertainty to handle the corresponding data. Therefore, we apply the following constraint called U-Loss:

$$L_u^m = \frac{1}{N} \sum_j^N \sigma_j^{m2} G_j(x^m). \quad (10)$$

With this constraint, during training, as the value of $\sigma_j^{m2}$ increases, the Gate assigns a smaller weight $G_j$ to the corresponding expert. Thus, when an expert's uncertainty is higher, its probability of being selected decreases.

Through the aforementioned process, we obtain clear features $h_i^m$ for each single modality. Combine features from different modalities by concatenating or adding them to create a joint feature $\hat{h}$, which can be utilized for downstream tasks. We employ a fully connected layer to predict the final result $\hat{y}$ and use Mean Squared Error loss as the loss function for this regression task,

$$L_{task} = \frac{1}{n} \sum_i^n (y - \hat{y})^2, \quad (11)$$

where $n$ represents the number of samples. Finally, we utilize $L_{total}$ as the overall loss function,

$$L_{total} = L_{task} + \alpha \sum_m^{m \in M} L_{kl}^m + \beta \sum_m^{m \in M} L_{aux}^m + \lambda \sum_m^{m \in M} L_u^m, \quad (12)$$

where $\alpha$, $\beta$ and $\lambda$ are trade-off hyper-parameter, as introduced in Section 4.2.

# 4 EXPERIMENTS

## 4.1 Datasets

**Dataset.** To verify the effectiveness of our proposed model in face of different kinds of noise scenarios i.e., modality missing and modalities' noise, we conduct MSA experiments on three multimodal datasets, namely CMU-MOSI [31] and CMU-MOSEI [32] for modality missing, and MVSA-single [33] for modalities' noise. In addition to that, we also conduct an extensive experiment on NYU Depth V2 [34] in order to prove EUAR's generalizability of real world's modalities. **CMU-MOSI** consists of 2,199 short monologue video clips, among those 1,284,229, and 686 samples are used as training, validation, and testing set. **CMU-MOSEI** contains 22,856 samples of movie review video clips from YouTube. Followed previous studies, 16,326 samples are used for training, 1,871 and 4,659 samples are used for validation and testing. Both datasets above are manually annotated in continuous sentiment scores between [-3,3], representing sentiment including *highly negative, negative, weakly negative, neutral, weakly positive, and highly positive*. **MVSA-Single** sentiment analysis dataset includes a set of image-text pair with manual annotations collected from social meida. The dataset can be utilized as a valuable benchmark for both single-view and multi-view sentiment analysis. **NYU Depth V2** is a public indoor scenes datasets, which are composed of two modalities, i.e., RGB and depth images. In this dataset, each object is labeled with a class and an instance number (cup1, cup2, cup3, etc). Following previous work [9], we also adopt the commonly used 9 out of the 27 scene categories and the remaining categories as "Others".

**Table 1: Comparisons with recent state-of-the-art MSA methods on the CMU-MOSI, CMU-MOSEI, NYU Depth V2 and MVSA-Single datasets.**

| Method | CMU-MOSI | | | CMU-MOSEI | | |
|---|---|---|---|---|---|---|
| | Acc7 | Acc2 | F1 | Acc7 | Acc2 | F1 |
| CubeMLP [2](2022) | 45.5 | 85.6 | 85.5 | **54.9** | 85.1 | 84.5 |
| MHMF-BERT [27](2022) | - | 85.3 | 85.3 | - | 85.6 | 85.6 |
| GCNet [11](2022) | 44.9 | 85.1 | 85.1 | 51.5 | 85.2 | 85.2 |
| MSG-MBA [4](2023) | 45.3 | 85.7 | 85.6 | 52.8 | 85.4 | 85.4 |
| ConFEDE [6](2023) | 42.3 | 85.5 | 85.5 | **54.9** | 85.8 | 85.8 |
| DiCMoR [7](2023) | 45.3 | 85.6 | 85.6 | 53.4 | 85.1 | 85.1 |
| **EUAR(Ours)** | **46.1** | **86.3** | **86.3** | **54.9** | **86.6** | **86.4** |

| Method | NYU-Depth V2 | | MVSA-Single | |
|---|---|---|---|---|
| | Acc | F1 | Acc | F1 |
| Concat | 70.30 | 69.82 | 65.59 | 65.43 |
| Late Fusion | 69.14 | 68.32 | 76.88 | 75.72 |
| MMTM [28] (2020) | 71.04 | - | 75.19 | 74.97 |
| TMC [29] (2021) | 71.06 | 69.83 | 76.06 | 74.55 |
| QMF [9] (2023) | 70.09 | 68.65 | 78.07 | 77.18 |
| MVCN [30] (2023) | - | - | 76.06 | 74.55 |
| **EUAR (Ours)** | **71.71** | **70.67** | **79.58** | **78.04** |

## 4.2 Implementation Details

**Evaluation Tasks.** We evaluate our model on multimodal sentiment analysis task, which aims to classify different sentiments into categories such as positive, negative, or neutral using bimodal or tri-modal data, including audio and visual data. Corresponding to our initiation, we perform the MSA task under the following noisy scenarios: *(1) Modality Missing:* Followed previous research on multimodal incompleteness, we employ *fixed missing strategy.* For this strategy, missing modalities are consistent for all samples, which means all samples have the same available modalities. Since there are three modalities: text, vision and acoustics, in CMU-MOSI and CMU-MOSEI dataset, seven distinct combinations of missing patterns are utilized in the experiments. We evaluate the performance using the metrics below for modality missing scenario: 7-class accuracy ($ACC_7$), binary accuracy ($ACC_2$) and F1 score. *(2) Modality Noise.* For this scenario, we add Salt and Pepper noise and Gaussian noise of different intensity to the visual modality; as for text modality, blank which replace the content is employed in the text in different ratio to simulate sample-wise noise. Finally, report the performance using following metrics: binary accuracy ($ACC_2$) and F1 score.

**Experiment Setup.** In alignment with alternative methodologies, we employ FACET, COVAREP and BERT [35] as feature extractors for visual, audio and text modalities in both CMU-MOSI and CMU-MOSEI. For bimodal dataset, we utilize ResNet-152 as the feature extractors for RGB and Depth images, accompanied with BERT as the texual feature extractors. The hyperparameters we employed, denoted as $\alpha$, $\beta$, $\lambda$, were set to 1e-3, 1e-5, and 1e-3, respectively. Additionally, for the selection of the number of experts using the Top-k mechanism, we set the value of k to 3. Due to space constraints, further relevant details are provided in the supplementary materials. In short, We implemented all the experiments using PyTorch on a RTX 3090 GPU with 24GB memory. We set the training batch size as 16 and train our model for 100 epochs. We run each experiments on the testing set and report the model's performance.

## 5 RESULTS AND ANALYSIS

To demonstrate the superiority of our approach, we conducted comparisons with state-of-the-art methods on three widely-used multimodal emotion analysis datasets. This includes two tri-modal video datasets, CMU-MOSI and CMU-MOSEI. The methods compared include CubeMLP [2], MHMF-BERT [27], GCNet [11], MSG-MBA

[4], ConFEDE [6] and DiCMoR [7]. In addition, it also includes a bi-modal text-image dataset, MVSA-Single. The methods compared include MMTM [28], TMC [29] and QMF [9]. Furthermore, we tested the performance of our model in situations of modality missing and noisy data.

## 5.1 Overall Comparisons

We report the comparison results of our method with the current state-of-the-art methods on datasets CMU-MOSI, CMU-MOSEI, and MVSA-Single in Table 1. From the table, it is evident that our method outperforms the current state-of-the-art methods in the majority of metrics, with a significant improvement. It is worth noting that our method achieves Acc2 and F1 metrics exceeding 86% on the CMU-MOSI dataset, pushing the model's performance to a new peak on this dataset. Additionally, our method also demonstrates outstanding performance on the CMU-MOSEI dataset, with Acc2 and F1 metrics surpassing the second closest by more than 1%. Our method also performs exceptionally well on the bimodal text-image classification dataset. It surpasses the current state-of-the-art methods by more than 1% in both Accuracy and F1 score metrics. At the same time, it is noticeable that our method does not outperform other methods on the Acc7 metric on the CMU-MOSEI dataset. We speculate that the high quality of the CMU-MOSEI dataset itself might have limited the advantage of our uncertainty routing method, resulting in slightly lagging performance compared to other methods.

## 5.2 Additional Experiments

In addition to analyzing the aforementioned three widely used sentiment analysis datasets, we conducted further experiments. Specifically, we conducted additional validation on the scene recognition dataset NYU Depth V2. The sentiment analysis datasets we used encompass features from three modalities: text, audio, and vision. However, the real world consists of more than just these three modalities of data, hence we decided to conduct more experiments. The NYU Depth V2 dataset for scene recognition includes features from two modalities: RGB images and depth images. By utilizing inputs from both modalities, we aim to recognize the scenes depicted in the images. We report the experimental results in the right subtale of Table 1. As can be observed from the graph, our approach still outperforms existing state-of-the-art methods even when including RGB images and depth images data. It is noteworthy that

**Table 2: Comparison on fixed missing strategy. The term "Available" denotes the presented modality. We rendered inactive all modalities except for the one indicated as "Available," evaluated the model's performance, and compared it with the current state-of-the-art methods. The values reported in each cell denote Acc2/F1/ACC7.**

| Datasets | Available | DCCA [36] | MCTN [37] | MMIN [38] | GCNet [11] | DiCMoR [7] | EUAR(Ours) |
|---|---|---|---|---|---|---|---|
| CMU-MOSI | { L } | 76.4/76.5/28.3 | 79.1/70.2/41.0 | 83.8/83.8/41.6 | 83.7/83.6/42.3 | 84.5/84.4/44.3 | **86.0/86.0/46.1** |
| | { V } | 52.6/51.1/17.1 | 55.0/54.4/16.3 | 57.0/54.0/15.5 | 56.1/55.7/16.9 | 62.2/60.2/20.9 | **64.9/64.9/23.6** |
| | { A } | 48.8/42.1/16.9 | 56.1/54.5/16.5 | 55.3/51.5/15.5 | 56.1/54.5/16.6 | 60.5/60.8/20.9 | **63.0/62.3/23.2** |
| | { L, V } | 76.7/76.8/30.0 | 81.1/81.2/42.1 | 83.8/83.9/42.0 | 84.3/84.2/43.4 | 85.5/85.4/45.2 | **86.2/86.2/45.5** |
| | { L, A } | 77.0/77.2/30.2 | 81.0/81.0/43.2 | 84.0/84.0/42.3 | 84.3/84.2/43.4 | 85.5/85.5/44.6 | **86.1/86.1/44.7** |
| | { V, A } | 54.0/52.5/17.4 | 57.5/57.4/16.8 | 60.4/58.5/19.5 | 62.0/61.9/17.2 | 64.0/63.5/21.9 | **66.1/65.8/24.2** |
| | { L, V, A } | 77.3/77.4/31.2 | 81.4/81.5/43.4 | 84.6/84.4/44.8 | 85.2/85.1/44.9 | 85.6/85.6/45.3 | **86.3/86.3/46.1** |
| CMU-MOSEI | { L } | 79.7/79.5/47.0 | 82.6/82.8/50.2 | 82.3/82.4/51.4 | 83.0/83.2/51.2 | 84.2/84.3/52.4 | **85.3/85.2/52.9** |
| | { V } | 61.1/57.2/40.1 | 62.6/57.1/41.6 | 59.3/60.0/40.7 | 61.9/61.6/41.7 | 63.6/63.6/42.0 | **66.3/65.3/42.4** |
| | { A } | 61.4/53.8/40.9 | 62.7/54.5/41.4 | 58.9/59.5/40.4 | 60.2/60.3/41.1 | 62.9/60.4/41.4 | **64.5/60.7/41.6** |
| | { L, V } | 80.4/80.4/47.1 | 83.2/83.2/50.4 | 83.8/83.4/51.2 | 84.3/84.4/51.1 | 84.9/84.9/53.0 | **86.0/86.0/53.2** |
| | { L, A } | 80.0/80.0/47.4 | 83.5/83.3/50.7 | 83.7/83.3/52.0 | 84.3/84.4/51.3 | 85.0/84.9/52.7 | **85.1/85.1/53.7** |
| | { V, A } | 62.7/59.2/41.6 | 63.7/62.7/42.1 | 63.5/61.9/41.8 | 64.1/57.2/42.0 | 65.2/64.4/42.4 | **66.3/65.3/42.7** |
| | { L, V, A } | 81.2/81.2/48.2 | 84.2/84.2/51.2 | 84.3/84.2/52.4 | 85.2/85.1/51.5 | 85.1/85.1/51.4 | **86.6/86.4/54.9** |

our method exceeds the state-of-the-art methods by 1% in accuracy, demonstrating a significant performance improvement. Additionally, our method performs remarkably well in terms of F1 Score. This proves the generalizability of our approach to applications beyond just text, audio, and visual modalities.

**Table 3: Comparisons with state-of-the-arts concerning model performance on noisy NYU Depth V2 datasets.**

| Method | Clean | Salt-Pepper Noise | | Gaussian Noise | |
|---|---|---|---|---|---|
| | $\epsilon = 0$ | $\epsilon = 5$ | $\epsilon = 10$ | $\epsilon = 5$ | $\epsilon = 10$ |
| Concat | 70.44 | 57.98 | 44.51 | 59.97 | 53.20 |
| Late Fusion | 69.16 | 56.27 | 41.22 | 59.63 | 51.99 |
| Align | 70.31 | 57.54 | 43.01 | 59.47 | 51.74 |
| MMTM | 71.04 | 59.45 | 44.59 | 60.37 | 52.28 |
| TMC [29] | 71.06 | 59.34 | 44.65 | 61.04 | 53.36 |
| QMF [9] | 70.09 | 58.50 | 45.69 | 61.62 | 55.60 |
| **EUAR (Ours)** | **71.71** | **61.35** | **46.63** | **63.15** | **57.79** |

## 5.3 Experiments on Noisy Datasets

In order to demonstrate the supremacy of our method in handling noisy data, we conducted experiments under scenarios of modality missing, Gaussian noise, Salt and Pepper noise, etc. The experimental results are reported in Tables 2 and 3. From Table 2, it can be observed that our method exhibits significant performance improvement compared to other methods under the modality missing condition. It is noteworthy that our method consistently outperforms the state-of-the-art methods by nearly 1% in all scenarios. Particularly, when considering only the visual modality, our method surpasses the second-best by nearly 3% across three metrics, demonstrating the advantage of our approach. Similarly, as only the acoustic modality is available, our EUAR outperforms the second-best by around 2%. In Table 3, we report the performance of our method under the

**Table 4: Ablation studies on training objectives on the CMU-MOSI and CMU-MOSEI datasets.**

| KL-Loss | U-Loss | CMU-MOSI | | | CMU-MOSEI | | |
|---|---|---|---|---|---|---|---|
| | | Acc2 | F1 | Acc7 | Acc2 | F1 | Acc7 |
| ✓ | ✓ | 86.3 | 86.3 | 46.1 | 86.6 | 86.4 | 54.9 |
| - | ✓ | 85.1 | 85.1 | 46.1 | 85.5 | 85.6 | 52.5 |
| ✓ | - | 83.8 | 83.7 | 44.1 | 85.3 | 85.2 | 51.5 |
| - | - | 81.7 | 81.6 | 31.1 | 85.2 | 84.8 | 50.7 |

presence of Gaussian noise and Salt and Pepper noise. It is evident that our method achieves state-of-the-art performance under both noise conditions. Specifically, our method shows a noticeable lead, surpassing the second-best by 2% when the Salt and Pepper noise intensity is 5 and the Gaussian noise intensity is 10. Whether in the case of modality missing or Gaussian and Salt and Pepper noise, our method consistently demonstrates superior performance in noisy data scenarios.

## 5.4 Further Analysis

**Ablation Study on Loss.** In order to investigate the effectiveness of the loss function proposed in our routing strategy, we conducted thorough ablation experiments on the loss function on both CMU-MOSI and CMU-MOSEI. We performed ablation experiments on the uncertainty loss $L_{kl}$ and the uncertainty routing loss $L_u$ separately, and the experimental results are shown in Table 4. From the table, it can be observed that the performance of the model significantly decreases on all metrics when any of the loss is ablated. It is worth noting that when both losses are ablated, compared to the complete model, every metric of the model's result decreases dramatically, especially on CMU-MOSI by more than 4%. The $L_{kl}$ loss enables better quantification of uncertainty by experts, while the $L_u$ loss helps the gate to route samples to the expert with lower uncertainty for processing. Especially, when we solely dissolve the KL-loss, our

performance only declines marginally, by merely around 1% on both datasets. This further corroborates the validity of our introduction of MoE. The intrinsic dynamism of MoE confers an advantage in this task, beyond solely relying on more precise routing methods. However,the improvements we propose complement the inherent nature of MoE. Only when these two losses interact can the overall effectiveness of the model be optimized.

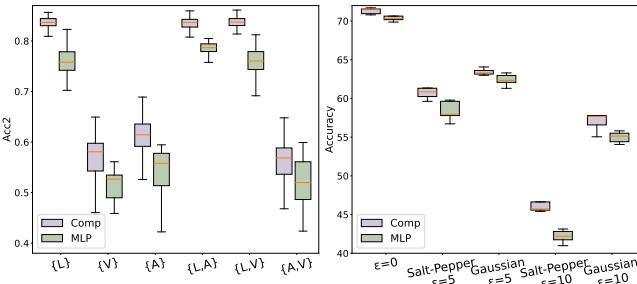

Figure 3: Analysis under different noise conditions. We substituted the "expert" with a regular MLP and tested the performance under varying noise conditions.

**Analysis on Model Robustness**. Since our motivation is to address noise within modalities, we conduct more ablation studies on the CMU-MOSI and MVSA-Single under different noise scenarios to verify the robustness and stability of our proposed model. In particular, on CMU-MOSI, we implemented different kinds of missing modality combinations, the experimental results are shown in the Figure 3. The MLP model which uses simple linear layers to process data (we use MLP model to represent this ablated model in the following paragraph as well) is adapted as the ablated model in the experiment under six distinct missing combinations. By observing the final results, it is obvious to find: the simple MLP model is outperformed by complete model on all missing combinations. Especially, when the text modality present in the combination, complete model possesses more evident performance. Contrastively, the complete model has more concentrated outcomes and reduced fluctuations, revealing its remarkable robustness and stability.

Moreover, on NYU Depth v2, we also replaced the expert with a standard MLP and tested under four different noise conditions, as shown in right picture of Figure 3. From the graph, it can be observed that our comprehensive approach significantly outperforms. It is worth noting that as the noise intensity increases, the advantage of our method becomes more apparent, demonstrating the excellence of our approach. Especially in the case of Gaussian noise with an intensity of 5, our method consistently outperforms the standard MLP in both average performance and stability.

**Ablation Study on Experts Number.** To further explore the impact of different experts number on our proposed MoE module, we conduct ablation experiment on experts number, as shown in Fig. 4. The numbers of experts range from 2 to 32, and it is clear to tell that as the number of expert increase, the model's performance present upward trend. The results indicate that with more experts, our model is able to exclude the noise in the samples more precisely,

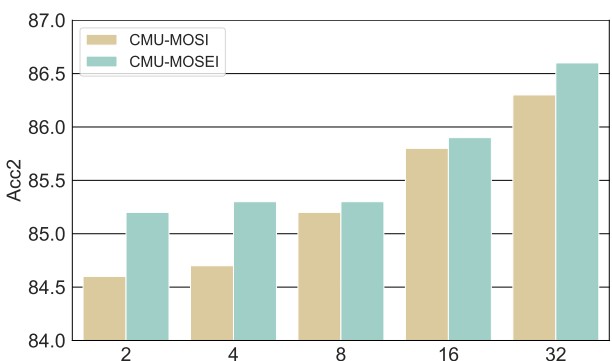

Figure 4: Ablation Study on Expert Number.We conducted experiments on the CMU-MOSI and CMU-MOSEI datasets, testing expert quantities ranging from 2 to 32, and reported the accuracy Acc2 metric.

and more discriminatively extract the useful features that are benefit the final classification results. It is noteworthy that when we increase the number of experts from 2 to 32, the performance of our model on Acc2 improves by nearly 2%. This demonstrates the success of our proposed dynamic network strategy. However, it can be observed that the improvements in Acc2 metrics on CMU-MOSEI during the early part are not as significant when changing the number of experts. We analyzed that the CMU-MOSEI dataset has a larger volume of data, requiring a greater number of parameters for fitting. This is also why there is a significant increase in Acc2 during the latter part of the experiment. With an increase in the number of experts in our approach, there is an enhancement in our network's ability to perform finer-grained sentiment analysis.

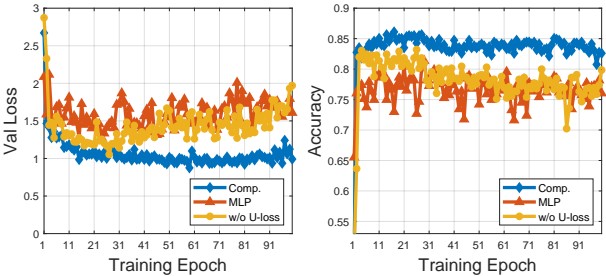

Figure 5: Analysis concerning the training convergence. We ablate different modules and observe performance fluctuations in training.

**Analysis on Training Process.** As illustrated in Fig 5, we also studied the our proposed model's training convergence and performance fluctuations in the training process on the CMU-MOSI. According to the results, it can tell that, different from the other two structural ablated model, the complete model exhibits a smoother training process and eventually obtained a remarkably better performance. Specifically, the training processes of two ablated models show that, compared to the MLP model, the other model that adapts our MoE structure without the proposed $L_u$ loss (U-loss in the legend) exhibits relatively better training convergence. The reason lies

in the MoE's capacity of quantitatively acquiring uncertainty of the projected features, forcing the ablated model to neglect the features with high uncertainty. On the other hand, due to the absence of proposed $L_u$ loss, the ablated model failed to route features to the experts more precisely, resulting in more fluctuating training process and worse final performance. In another word, proposed U-loss can help assist the MoE structure to adaptively select experts based on uncertainty, acquiring more steady training processes and more salient results. In a conclusion, the structural ablation experiments above proves again the rationality and effectiveness of our proposed model.

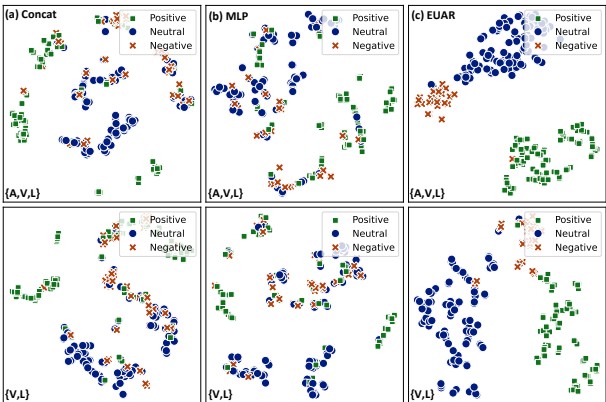

Figure 6: Visualizations of joint representations generated by different abalted models with t-SNE [39] on the CMU-MOSI dataset. The upper row represents the results under the full modality condition, while the lower row depicts the results when the audio modality is absent.

**Visualizations of Joint Representations** Furthermore, using the features that are projected into a 2D space by t-SNE is a straightforward way to exhibit learning joint representations of EUAR. Hence, we employ the t-SNE to visualize the learned joint representations of simple concatenation, MLP, and complete EUAR for a quantitative comparison. In detail, we randomly select joint features processed by our proposed EUAR model from the testing set of the CMU-MOSI dataset, and use three colors to represent their true sentimental labels.

As illustrated in Fig 6, the picture above is the visualization of complete modalities combination, while the figure below is the combination whose acoustic modality is unavailable. It is evident to observe that the features generated by the complete model are in compact and discriminative distribution. On the contrary, when MLP are utilized, the ablated model can separate the features in different classes to some degree, the final distribution is less compact and distinguishing, resulting in corrupted classification results at last. And the clusters are still closer than those generated by the complete EUAR, indicating less discriminability. Consequently, the t-SNE visualization results indicate that with the help of both MoE structure and proposed U-loss, effectively enables our EUAR model to resist noise in the modalities, making it more capable of learning a representative joint representations. Because of the limitation of the

space, more missing modality combinations' t-SNE visualizations will be attached in the supplementary materials.

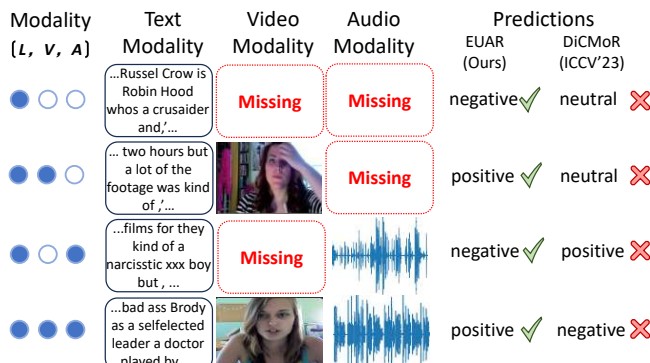

Figure 7: Visualizations of test cases selected from the CMU-MOSI datasets. It can be observed that our EUAR method reveals better robustness against the noise.

**Qualitative Analysis** Additionally, for a qualitative study, we also illustrate several representative test cases from CMU-MOSI with different missing modalities situations. Specially, when the modality is unavailable in the combination, we use rectangles with red dash lines to denote. As shown in Fig 7, we compare our proposed model with the recent counterpart DiCMoR [7]. Under different missing modalities circumstances, our EUAR method consistently produce correct classifications, while DiCMoR fails to fulfill the task accurately. It explicates that with the assistance of the MoE architecture and the novel routing algorithm we proposed, our model is capable of learning more stable and robust joint representations from deteriorated multimodal data and excluding the interference of noise that exists across the modalities, gaining noise-resistant performance with higher accuracy.

## 6 CONCLUSION

In this paper, we proposed a novel multimodal sentiment analysis framework called Enhanced Experts with Uncertainty-Aware Routing (EUAR), which excelled at handling noise in multimodal data and dynamically adjusted the network based on different samples. Specifically, we introduced the MoE in multimodal sentiment analysis tasks to address varying noise levels in multimodal data. Particularly, we enhanced the functionality of experts to quantify uncertainty and extract clear unimodal features. Moreover, we devised novel routing strategies to train the model to route samples with different noise levels to corresponding experts with lower uncertainty for processing. Our approach outperformed existing state-of-the-art methods in multimodal sentiment analysis tasks and excelled in extended experiments with additional modalities. Furthermore, we conducted experiments under conditions of missing modalities and noisy data, demonstrating the superiority of our method in handling noisy data. For future work, we aim to delve deeper into methods for handling noise in multimodal data.

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
