# OpenReview forum: "Enhanced Experts with Uncertainty-Aware Routing for Multimodal Sentiment Analysis"
_acmmm.org/ACMMM/2024/Conference — MM2024 Poster_

### Official Review · Reviewer_UbmL · 2024-05-19

**Rating:** 3
**Confidence:** 3

**Summary:**

This paper focuses on multimodal sentiment analysis, and proposes a EUAR model, i.e. Enhanced experts with Uncertainty-Aware Routing, to address the influence of noisy data on multimodal sentiment analysis. EUAR can capture uncertainty by quantifying noise in samples to obtain clearer features and utilize this uncertainty to guide the routing algorithm according to different noise scenarios. This paper shows the advantages of EUAR comparing with related work by experiments.

**Strengths:**

1. This paper proposes Uncertainty-Aware Routing, which introduces uncertainty to guide the routing algorithm. This is an active learning idea, and it should be novel for sentiment analysis.
2. The experiment setup is described clearly by this paper, and the experimental process is complete and includes comparison with other work, selection of parameters, ablation, and so on.
3. This paper makes full use of multimedia technique to show the experimental results and gives detail analysis.

**Limitations:**

About the details and process of EUAR model, the description of the model in this paper is not very clear, which greatly affects the readability and understandability of the paper. Main problems are following.
1. According to Figure 1, this paper classifies noise data into modal-wise noise and sample-wise. However, what is the meaning and function of this classification? How does EUAR model treat these two types of noise differently?
2. As the main framework, figure 2 is not clear enough, or does not have a clear correspondence with the paper text. In the text, visual input can be extracted audio, visual, and text feature, what for text input? In Figure 2, for Enhanced Experts module, only the process of text can be seen. The paper labels (Textual/Visual) in the figure, but just replace T with V? In addition, Figure 2 should be more detailed, and some of the variables in the text should be marked more in the figure.
3. This paper classifies experiment datasets into modality missing and modality noise. On the one hand, how does this classification relate to the previous modal-wise noise and sample-wise? On the other hand, CMU-MUSI, CMU-MUSEI, and MVSA-single are multimodal sentiment analysis datasets, how does this paper process them to be suitable for modality missing and modality noise experiments? Do the 3 datasets themselves have modality missing and modality noise problems? If yes, the paper should give the statistical information of 3 datasets.

**Suitability:**

3

---

### Official Review · Reviewer_mnkP · 2024-05-21

**Rating:** 4
**Confidence:** 4

**Summary:**

This paper introduces the Enhanced experts with Uncertainty-Aware Routing (EUAR) method to mitigate the impact of noisy data on MSA. Leveraging a Mixture of Experts approach, EUAR dynamically adjusts the network based on data uncertainty, refining features and routing mechanisms. Through the proposed U-loss, the network routes samples to experts with lower uncertainty, yielding clearer features. Experimental results, including on noisy datasets, showcase EUAR's superiority, establishing new benchmarks in multimodal sentiment analysis.

**Strengths:**

1. The motivation for this study is relatively novel within the field, addressing the pressing need to mitigate the impact of noisy data on multimodal sentiment analysis. Data sourced from social media platforms often contain significant levels of noise, presenting a unique challenge for accurate analysis.

2. The manuscript is commendably readable and easy to follow, facilitating understanding and comprehension of the presented research.

3. The proposed method achieves state-of-the-art (SOTA) performance on selected datasets, indicating its effectiveness and competitiveness in the field of multimodal sentiment analysis. Additionally, the experiments conducted are comprehensive, covering various aspects necessary for thorough validation.

**Limitations:**

1. In line 339, the phrase "being perturbed by noise 𝜎 to obtain𝑧" should include a space between "obtain" and "𝑧." Manuscripts submitted to top-tier conferences should avoid such low-level errors to maintain a high standard of professionalism and accuracy. Attention to detail in formatting and presentation is crucial for ensuring the credibility and quality of academic research.

2. This study places excessive emphasis on capturing data uncertainty, while insufficiently addressing the extraction of sentiment clues from different modalities. This focus detracts from the task-specific objectives of multimodal sentiment analysis, where the integration and interpretation of emotional cues across various data types are crucial. A more balanced approach that equally prioritizes sentiment extraction from all modalities would enhance the method's relevance and effectiveness for the intended task.


3. While the design of the algorithms is reasonable and effective, the presentation of the methodology may give readers the impression that it primarily addresses noise reduction in multimodal scenarios rather than focusing on multimodal sentiment analysis. Clarifying how the proposed method specifically enhances sentiment analysis, beyond merely mitigating noise, would strengthen the relevance and impact of the research within its intended context.

4. The concept of "expert" in this study is too abstract. The paper lacks clarity regarding the specific types of sub-models or neural networks chosen as experts within the framework. Providing detailed information about the architecture and characteristics of these experts would enhance the transparency and comprehensibility of the proposed method.

5. The overall loss function includes too many hyperparameters, making it difficult to determine whether all these components are beneficial for the task.

**Suitability:**

3

---

### Official Review · Reviewer_EL4Z · 2024-05-23

**Rating:** 4
**Confidence:** 3

**Summary:**

This paper applies the MoE (Mixture of Experts) framework in multimodal sentiment analysis. This involves dynamically adjusting the network to mitigate noise within individual modalities and across diverse samples. They develop a routing algorithm to bolster the capabilities of the individual experts within the network. This algorithm enables the network to assess the uncertainty associated with noise and leverage this information to guide the selection of experts with greater confidence for processing each sample. Their experiments have resulted in achieving state-of-the-art performance levels. Moreover, their method has showcased superior robustness in handling noisy conditions compared to existing techniques.

**Strengths:**

The authors introduce a pioneering approach by formulating a MoE framework in MSA, considering multiple modalities. The proposed framework presents a novel method to enhance the robustness capabilities of MSA models. The empirical results demonstrate the superior performance of the proposed method.

**Limitations:**

1. Traditional methods such as TFN are missing as baselines, and classic MSA methods like Self-MM, MAG-BERT, and MISA are also absent.
2. Additionally, there is another piece of work addressing the generalization and robustness issues in MSA. The paper "General Debiasing for Multimodal Sentiment Analysis" published at ACM MM 2023 should be discussed to highlight the differences in approach compared to this work.

**Suitability:**

3

---

### Meta-Review · Program_Chairs · 2024-07-13

**Recommendation:** Accept (Poster)
**Confidence:** 4

**Metareview:**

This paper proposes a new method, Enhanced experts with Uncertainty-Aware Routing (EUAR), to better mitigate noisy date by dynamically adjusting the network for improving multimodal sentiment analysis. All the reviewers highlight the novelty of the approach and its effectiveness, advancing the state of the art. They all align towards acceptance.